# Engaging community members in setting priorities for nutrition interventions in rural northern Ghana

**Maxwell Ayindenaba Dalaba** [1,2]* , **Engelbert A. Nonterah** [1,3]* , **Samuel T. Chatio**[1], **James K. Adoctor**[1], **Edith Dambayi**[1], **Esmond W. Nonterah**[1], **Stephen Azalia**[1], **Doreen Ayi-Bisah**[1], **Agnes Erzse**[4], **Daniella Watson**[5], **Polly Hardy-Johnson**[6,7], **Sarah H. Kehoe**[5,6], **Aviva Tugendhaft**[4], **Kate Ward**[5,6], **Cornelius Debpuur**[1,8], **Abraham Oduro**[1,8], **Winfred Ofosu**[9], **Marion Danis**[10], **Mary Barker**[5,6,11,12], on behalf of the INPreP study group[¶]

1 Navrongo Health Research Centre, Ghana Health Service, Navrongo, Ghana, 2 Institute of Health Research, University of Health and Allied Sciences, Ho, Volta Region, Ghana, 3 Julius Global Health, Julius Centre for Health Science and Primary Care, University Medical Centre Utrecht, Utrecht University, Utrecht, The Netherlands, 4 SAMRC/ Wits Centre for Health Economics and Decision Science, PRICELESS SA, University of Witwatersrand School of Public Health, Faculty of Health Sciences, Johannesburg, South Africa, 5 Global Health Research Institute, School of Human Development and Health, Faculty of Medicine, University of Southampton, Southampton, United Kingdom, 6 Medical Research Council Lifecourse Epidemiology Centre, University of Southampton, Southampton, United Kingdom, 7 AECC University College, Bournemouth, United Kingdom, 8 Research and Development Division, Ghana Health Service, Accra, Ghana, 9 Upper East Regional Health Directorate, Ghana Health Service, Bolgatanga, Ghana, 10 Department of Bioethics, National Institutes of Health, Bethesda, MD, United States of America, 11 NIHR Southampton Biomedical Research Centre, University Hospitals Southampton Foundation Trust, Southampton General Hospital, Southampton, United Kingdom, 12 School of Health Sciences, University of Southampton, Highfield, Southampton, United Kingdom

☙ These authors contributed equally to this work.
¶ Membership of the INPreP study group is listed in the Acknowledgments.
* madalaba@yahoo.com, mdalaba@uhas.edu.gh (MAD); drenanonterah@gmail.com (EAN)

**Data Availability Statement:** All relevant data are within the paper or Supporting Information files.

**Funding:** This research was commissioned by the National Institute for Health Research (NIHR) for

## Abstract

This study used "Choosing All Together" (CHAT), a deliberative engagement tool to prioritise nutrition interventions and to understand reasons for intervention choices of a rural community in northern Ghana. The study took an exploratory cross-sectional design and used a mixed method approach to collect data between December 2020 and February 2021. Eleven nutrition interventions were identified through policy reviews, interaction with different stakeholders and focus group discussions with community members. These interventions were costed for a modified CHAT tool—a board-like game with interventions represented by colour coded pies and the cost of the interventions represented by sticker holes. Supported by trained facilitators, six community groups used the tool to prioritise interventions. Discussions were audio-recoded, transcribed and thematically analysed. The participants prioritised both nutrition-sensitive and nutrition-specific interventions, reflecting the extent of poverty in the study districts and the direct and immediate benefits derived from nutrition-specific interventions. The prioritised interventions involved livelihood empowerment, because they would create an enabling environment for all-year-round agricultural output, leading to improved food security and income for farmers. Another nutrition-sensitive, education-related priority intervention was male involvement in food and nutrition

the NIHR Global Health Research Group on leveraging improved nutrition preconception, during pregnancy and postpartum in sub-Saharan Africa through novel intervention models, Southampton 1000 DaysPlus Global Nutrition, at the University of Southampton, through UK Official Development Assistance (ODA) via the Department of Health and Social Care (Grant/Award Number: 17\63\154). The views expressed in this publication are those of the author(s) and not necessarily those of the NIHR or the Department of Health and Social Care. The funders had no role in study design, data collection and analysis, decision to publish, or preparation of the manuscript.

**Competing interests:** The authors have declared that no competing interests exist.

practices; as heads of household and main decision makers, men were believed to be in a position to optimise maternal and child nutrition. The prioritised nutrition-specific intervention was micronutrient supplementation. Despite low literacy, participants were able to use CHAT materials and work collectively to prioritize interventions. In conclusion, it is feasible to modify and use the CHAT tool in public deliberations to prioritize nutrition interventions in rural settings with low levels of literacy. These communities prioritised both nutrition-sensitive and nutrition-specific interventions. Attending to community derived nutrition priorities may improve the relevance and effectiveness of nutrition health policy, since these priorities reflect the context in which such policy is implemented.

## Introduction

In both low- and high-income countries, health services allocation decisions pose a challenge for health systems. Given that no health system around the world can afford to provide all possible services and treatments for its citizens, priority setting is essential [1]. When decisions to allocate resources and set priorities for healthcare and other health-related interventions are ineffective, limited resources are wasted and access to care is negatively affected [2].

Involving the community in these decision making processes has the potential to benefit policy makers as well as communities and foster community acceptance and ownership of an intervention or policy [3]. In recent times, decision makers are increasingly involving the community in priority setting for health services and policy making [3].

Priority setting for nutrition is no exception and decisions about which interventions to fund should reflect community values while considering the available budget. Community participation could be used to inform difficult decisions about resource allocation and therefore has the potential to improve the effectiveness of policy implementation [3, 4].

Involving potential beneficiaries in priority setting processes will enhance the relevance of nutrition interventions and is likely therefore to increase the likelihood of them being implemented. Inadequate or no community engagement might hinder intervention implementation and intended goals [5, 6]. In addition, when engaging communities, it is important to be as inclusive as possible and avoid exclusion of those with lower levels of education; their perspectives will be important in guaranteeing the value of decisions made and priorities set.

One approach to involving the community in priority setting is using community engagement or public deliberative tools such as Choosing All Together (CHAT). CHAT is a simulation decision-making tool that helps lay people understand that not all services can be provided, and that decisions need to be taken to identify services that are most important for them [3, 4, 7]. The CHAT tool is designed like a board game whereby interventions are represented by colour coded slices of the overall 'pie' and the cost of the interventions are represented by sticker holes. Participants engage with the board by distributing a limited number of stickers (representing the budget) on the board as they select the interventions that are most important to them. The stickers that are available to the participants are only able to cover approximately 60% of the sticker holes and therefore only some interventions can be selected [3].

CHAT was originally developed to design health insurance benefit packages with the public and this has been successfully modified and used in many countries such as India, South Africa, Switzerland and USA for health coverage decisions [3, 4, 7, 8]. For instance, CHAT was implemented in a rural community in South Africa to determine priorities for a health services package. The study demonstrated that deliberative engagement methods can be successful in helping

communities balance trade-offs and in eliciting social values around health priorities [3]. Similarly, CHAT was conducted among residents of Switzerland to discuss priorities for Swiss Health insurance coverage. Results showed that participants were able to engage with complex resource allocation trade-offs in healthcare coverage and to agree on a set of priorities [4].

CHAT has not yet been used by communities to prioritise nutrition interventions despite the growing appreciation of public engagement in priority-setting for health services. This is despite the fact that there are limited tools for facilitating deliberation in reaching consensus in the selection of health interventions [3].

This study therefore sought to use CHAT to identify priorities for nutrition intervention and to explore the reasons for their choices in rural communities in northern Ghana.

## Materials and methods

### Study area

The study was carried out in the Kassena-Nankana East and West Districts of the Upper East Region of Ghana. The two districts cover an area of about 1,674 square kilometres [9, 10]. The area is characterized by a rainy season from May to October and a prolonged dry season from October to March. Most people live on subsistence farming with the main crops being millet, rice, maize and groundnuts as well as animal rearing.

The study districts are located in the region with the highest level of malnutrition in the country. The prevalence of stunting (low height-for-age) and wasting (low weight-for-height) in the study region is 22.4% and 9.0% respectively [11].

### Study design

The study adopted an exploratory cross-sectional design and used mixed-methods to collect data between December 2020 and February, 2021. The study adapted the CHAT approach and used it to facilitate community group deliberations and identify how nutrition interventions were prioritized by community members in rural Ghana and the reasons for their choices.

This study was part of a project called Improved Nutrition during Preconception, Pregnancy and Post-delivery (INPreP). INPreP is a National Institute of Health Research (NIHR) funded global health research group coordinated by the University of Southampton, which seeks to engage with community and relevant stakeholders in optimising nutrition in the 1000 days plus period and to prioritise solutions for Ghana, Burkina Faso and South Africa [9, 12–15].

### Study population and sampling

For the identification of relevant policies and interventions, study participants constituted community members and stakeholders in health, nutrition and maternal and child health as well as non-governmental organizations. Stakeholders were identified and in-depth interviews (IDIs) carried out with them. Community members sampled from the study communities were scheduled and grouped for Focus Group Discussions (FGDs). Details of the sampling, recruitment and results have been published previously [9, 14, 15].

For the implementation of the CHAT tool, participants were recruited from the north and south zones of the Navrongo Health and Demographic Surveillance System (NHDSS) coverage area. The NHDSS database was used as the sampling frame for the selection of participants [10, 16]. Using the NHDSS database, a computerised simple random sampling method was used to select 6–12 participants from the south zone of the districts to represent the Nankana ethno-linguistic group and another 6–12 from the north zone to represent the Kasena ethno-linguistic group. The CHAT exercise was conducted in the respondents' respective

communities. Members of the research team visited the selected individuals and invited them to participate in the exercise. Out of the 6 deliberative group exercises that were conducted, 3 were with men and 3 with women who were aged 18 to 50 years, with each group having between 6 and12 participants (Table 1).

## Adapting and modifying CHAT tool for the Ghanaian context

We took five steps in the modification and implementation of the CHAT tool for nutrition interventions in this rural setting in northern Ghana. The CHAT tool was originally developed by the US National Institutes for Health and Michigan State University involving residents of North Carolina [3, 17], but a similar modification of CHAT had been done in a rural community in South Africa where it was used to determine priorities for a health services package [3]. The five steps to modify the tool are described below.

**Step 1: Desktop review to identify relevant nutrition interventions.** A desktop review was conducted of published and unpublished literature (web-based and hard copies) on nutrition and maternal and child health policies in Ghana. We considered a twenty year period, 1998 to 2018 and the search was chronological, contained details of the policies and the related evidence and whether the policy has been implemented or not [14]. The policy documents identified and reviewed are presented as (S1 Table).

**Step 2: Stakeholder and community engagement to identify relevant nutrition interventions.** We next identified and interviewed stakeholders with relevant knowledge of maternal and child nutrition issues and related policies and interventions. We developed a list of all potential stakeholders based on the literature search and used purposive sampling with snowballing to sample the stakeholders from agencies we identified as responsible for these policies [14]. We interviewed representatives at the local, national and sub-national level from government, civic society and United Nations (UN) agencies. In total, eleven IDIs were conducted and the interviewees provided us with additional sources of interventions, including unpublished policy documents. Interviewees were invited for a second round of engagement which comprised two FGDs, with an average of 12 individuals per group. Stakeholders were divided into two homogeneous groups based on the sector, operational level and organization of the participants [14]. Discussions aimed to identify and understand the current nutrition interventions, the needs and priorities in the various sectors.

Community engagement consisted of ten FGDs with men and women from communities in the study area to appreciate the community's view of the nutrition interventions that were important to them. These focus groups are reported in a previous publication which identifies a short list of context-specific maternal and child nutrition interventions [15].

**Step 3: Consolidating identified interventions.** The interventions identified in Steps 1 and 2 were consolidated, categorised and then used for the CHAT exercise [14, 15]. A total of 11 interventions were identified and grouped into 6 themes including community nutrition

**Table 1. Summary of community groups and study participants.**

| Group | Gender | Age group (years) | Ethnicity | Community |
|---|---|---|---|---|
| 1 | Women | 35–50 | Nankam | Azaasi-Kandiga |
| 2 | Women | 40–50 | Kassem | Manyoro |
| 3 | Men | 40–50 | Nankam | Bui |
| 4 | Men | 35–50 | Kassem | Nankolo |
| 5 | Women | 26–39 | Nankam | Bembisi-Kandiga |
| 6 | Men | 24–34 | Kassem | Janania |

education, youth education, male involvement, livelihood empowerment, health system strengthening and micronutrient supplementation (Fig 1). Of these 11 interventions two were nutrition specific (micronutrient supplementation and food fortification) and nine were nutrition sensitive (livelihood empowerment through sustainable agriculture, community nutrition education, youth education, health system strengthening and male involvement) (see Fig 1).

**Step 4: Costing exercise and development of the modified CHAT.** The 11 CHAT interventions that were identified in Step 3 were then costed. An ingredients approach where quantities of the resources required to deliver the intervention are multiplied by their unit price was used to estimate the cost of implementing the interventions in the community per year. All costs were estimated in local currency, Ghana cedis (GH¢) and results are presented in Ghana cedis (GH¢) and US dollars (US$). We used the average exchange rate for 2020 (1US$ = GH¢ 5.4). Unit prices were obtained from the open market and demographic information that describes the communities was obtained from the NHDSS database. The cost components included personnel costs, transportation costs to conduct interventions, cost of materials involved in the interventions and community mobilization costs.

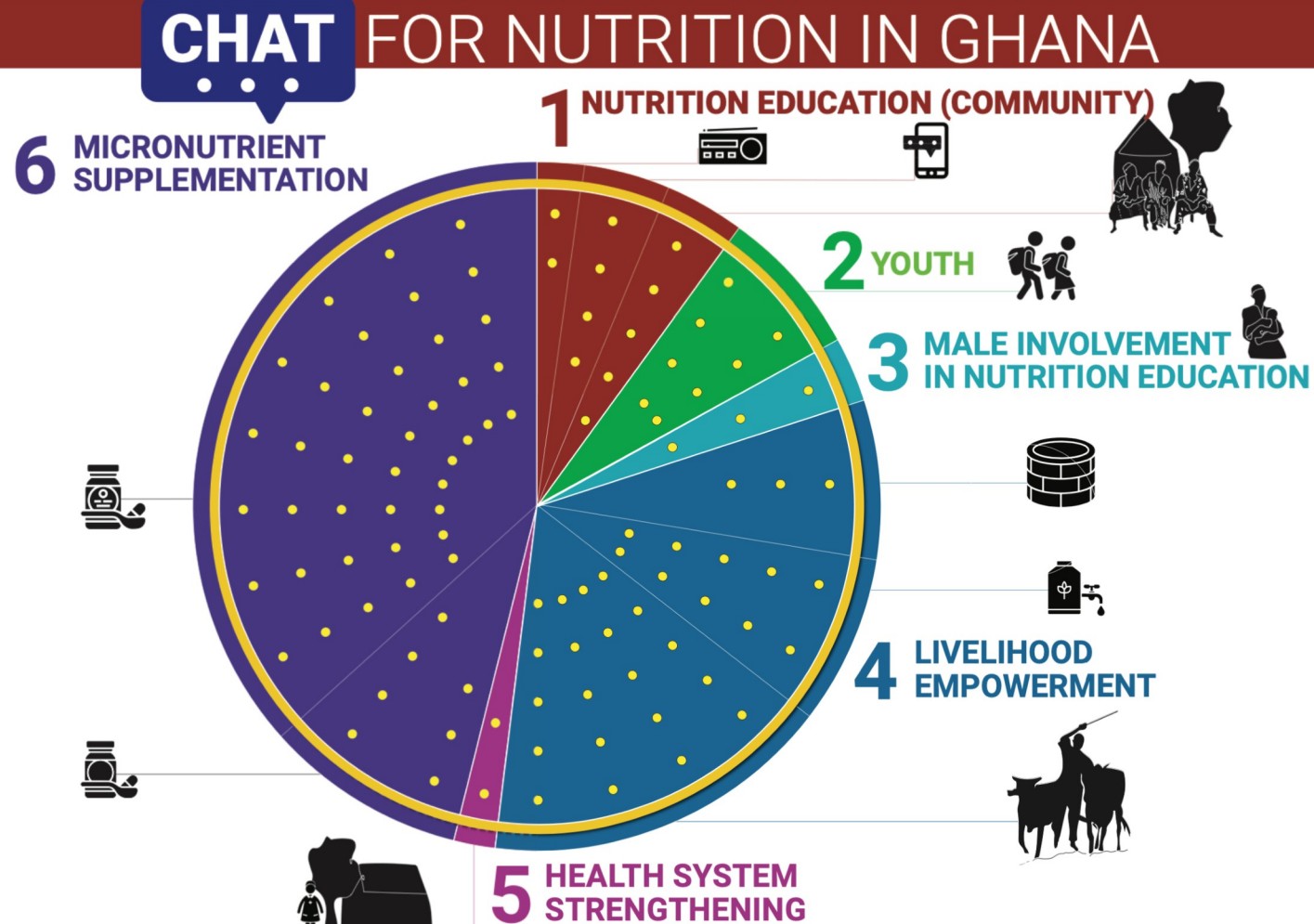

**Fig 1. The Navrongo, Ghana modified CHAT board used for the prioritization exercise.** 'The Navrongo, Ghana modified CHAT board used for the prioritization exercise,' Copyright 2005 The Regents of the University of Michigan, is licensed under Creative Commons Attribution 4.0 International Public License (CC BY 4.0) https://creativecommons.org/licenses/by/4.0/legalcode.

The list of the identified nutrition-related interventions and their associated costs that were included in the CHAT board and contained in the CHAT manual is presented as (S2 Table). The total cost of the package of interventions was estimated at GH¢652,980.00 / US $120,922.22.

The modified CHAT board, designed to fit the nutritional interventions specific to the study area, is presented in Fig 1. The CHAT board is divided into different colour coded 'pie' slices containing six broad nutrition areas with 11 listed interventions with different icons for each of the 11 interventions.

The cost of each intervention is represented by sticker holes as used in previous CHAT modifications [8]. Thus, each category of nutrition interventions had a specific cost depicted by the sticker holes. The total cost of the interventions was represented by 104 sticker holes across the whole board.

**Step 5: Conducting the CHAT exercise.**   The CHAT exercises were facilitated by six experienced research officers who were familiar with the study communities and specifically trained to deliver CHAT. Prior to implementation, the research officers received training on the research protocol including the CHAT facilitator script that provides step by step guidance on conducting the entire CHAT process (S1 Text). A pre-test was conducted during the training to determine the feasibility of conducting the CHAT process in these communities.

Key to implementing the CHAT exercise was the facilitator who directed participants through a series of rounds of deliberations in which participants were required to allocate a limited number of stickers–their budget—to interventions that they selected from the 11 on the CHAT board. Participants could test and understand the potential benefits of the selected interventions through hypothetical scenarios that were given to them by the facilitator. These scenarios were developed to illustrate the impact of giving priority to the 11 interventions within the constraints of the Ghana health-care system. Participants were encouraged to deliberate some of the consequences of either choosing or forgoing the nutrition programmes and reflect on the appropriateness of their choices (S2 Text).

Participants received 60 stickers which represented the units of currency/money they had available to allocate to the 104 sticker holes. These 60 stickers was able to cover approximately 58% (60/104) of the nutrition intervention options on the CHAT board. Giving a limited number of stickers enables decision making within the financial constraints of a budget and necessitates trade-offs between choices in a way that simulates the trade-offs that would have to be made in real decision-making processes. A 60% limit to selection of options for intervention is similar to the limits set for prior CHAT exercises [3]. With 60 stickers, each sticker represents 1/60th of the total amount of funds that can be used to prioritize the nutrition interventions listed on the CHAT board [3].

The various nutrition interventions were explained in detail in the CHAT manual (S2 Table) that accompanied the CHAT board. The entire exercise was conducted in the local language and the facilitators translated the manual during the exercise into either Kasem or Nankam. The information provided in the manual included the descriptions of the nutrition interventions, delivery mechanisms and the cost of the intervention which were represented in sticker value.

The following sequence was used in this CHAT exercise. Initially, participants went through the **first CHAT Round** during which they worked in pairs, and when they selected nutrition packages that they thought best for their community. Each of the participants received a **nutrition scenario card** that explained some of the consequences of the nutrition programmes. The **second CHAT Round** involved participants working together as an entire group, moderated by the facilitator, selected a nutrition package for the whole community. The groups were expected to reach an agreement about how to allocate the stickers through a

majority vote. The full deliberation process was audio-recorded with consent from participants and the entire exercise took about 3 hours for participants to complete.

## Ethics approval and consent to participate

Ethical approval for the study was obtained from the Navrongo Health Research Centre Institutional Review Board (Approval ID: NHRCIRB129). Written informed consent was obtained from all participants before data collection.

## Data analysis

Group deliberations were conducted in the local languages, audio recorded, translated and transcribed into English. For the qualitative analysis, transcripts were imported into a qualitative data analysis software programme (NVivo 12) which organized the data in preparation for thematic analysis. This qualitative work was carried out to understand the group choices, the reasons for the choices and the nature of the group deliberations.

The quantitative aspect of the research sought to obtain numerical information on the intervention choices using the stickers. It also provided support to the qualitative data on group choices on the interventions. The quantitative data which included the sticker choices of the interventions were captured on paper and then transferred to Microsoft Excel spread sheet for analysis.

## Results

### Socio-demographic characteristics of participants

A total of 53 community members in 6 groups participated in the CHAT exercise. There was an average of 8 participants per session. The average age was 39 years (range: 24–50 years). The majority of participants (53%) were women and over 90% were married. With regards to educational status, 37.7% had primary education and 26.4% had no formal education (Table 2).

**Table 2. Socio-demographic characteristics of the study participants.**

| Characteristics | Frequency(n = 53) | Percent (%) |
|---|---|---|
| **Age** | | |
| 24–30 | 9 | 17.0 |
| 31–40 | 19 | 35.9 |
| 40 and above | 25 | 47.1 |
| **Gender** | | |
| Women | 28 | 52.8 |
| Men | 25 | 47.2 |
| **Marital Status** | | |
| Married | 49 | 92.5 |
| Single | 3 | 5.7 |
| Widowed | 1 | 1.9 |
| **Education Level** | | |
| Junior High | 10 | 18.9 |
| No education | 14 | 26.4 |
| Primary | 20 | 37.7 |
| Senior High | 7 | 13.2 |
| Tertiary | 2 | 3.8 |

**Table 3. Final group choices for each nutrition intervention.**

| Intervention | Number of stickers on board | G1 (W35-50) | G2 (W40-50) | G3 (M 40–50) | G4 (M 35–50) | G5 (W26-39) | G6 (M24-34) | Total number of groups selecting a topic | % of stickers (intervention stickers/ total stickers) |
|---|---|---|---|---|---|---|---|---|---|
| *1. Nutrition education (community)* | | | | | | | | | |
| 1a.Radio broadcasting | 2 | 0 | 0 | 1 | 0 | 1 | 0 | 2 | 1.92 |
| 1b.SMS and posters. | 4 | 0 | 0 | 0 | 0 | 0 | 0 | 0 | 3.85 |
| 1c.Food demonstration in durbars* | 5 | 0 | 0 | 1 | 1 | 1 | 1 | 4 | 4.81 |
| *2. Youth* | | | | | | | | | |
| 2a.Out of school and in school youth nutrition education | 8 | 0 | 0 | 1 | 1 | 1 | 1 | 4 | 7.69 |
| *3. Male involvement in nutrition education* | | | | | | | | | |
| 3a.Male involvement in nutrition education | 3 | 0 | 1 | 1 | 1 | 1 | 1 | 5 | 2.88 |
| *4. Livelihood empowerment* | | | | | | | | | |
| 4a.Water wells and water tanks | 3 | 1 | 1 | 1 | 1 | 1 | 1 | 6 | 2.88 |
| 4b.Agricultural inputs | 10 | 1 | 1 | 1 | 1 | 1 | 1 | 6 | 9.62 |
| 4c.Livelihood skills training | 20 | 1 | 1 | 1 | 1 | 1 | 1 | 6 | 19.23 |
| *5. Health system strengthening* | | | | | | | | | |
| 5a.Health system strengthening | 2 | 0 | 0 | 0 | 1 | 0 | 0 | 1 | 1.92 |
| *6. Micronutrient supplementation* | | | | | | | | | |
| 6a.Iron-folate supplementation | 7 | 1 | 1 | 1 | 1 | 1 | 1 | 6 | 6.73 |
| 6b.Food fortification | 40 | 1 | 1 | 1 | 1 | 1 | 1 | 6 | 38.46 |
| | 104 | 5 | 7 | 9 | 9 | 9 | 8 | 47 | 100.00 |

G, group; W, women and M, men

*Community gathering involving chiefs, elders and community members to deliberate on a particular issue

## Final group choices for each nutrition intervention

Table 3 presents the final group choices for each nutrition intervention as well as percentage of stickers that were required to cover the cost of implementing each intervention. When an intervention was selected by a group, it was represented by "1", "0" indicating that an intervention was not selected. The results showed that there were some interventions that were not selected by some groups while other interventions were selected by all the groups. Though all the groups considered a variety of interventions, participants prioritized more nutrition specific interventions that have direct benefits over the nutrition sensitive education related interventions [18].

Interventions that were selected by all the groups included all three interventions under *livelihood empowerment* (water wells and water tanks, agricultural inputs, livelihood skills training) and both *micronutrient supplementation interventions (*iron-folate supplementation and food fortification). The nutrition education intervention that was not selected by any of the groups included SMS and posters, while radio broadcasting was selected by only 2 groups. Health system strengthening was considered a similarly low priority, with only one out of six groups selecting it. This group consisted of men aged between 35 and 40 years.

**Priority choices and the justification of choices.** *Livelihood empowerment.* All three of these interventions (water wells and water tanks, agricultural inputs, and livelihood skills

training) were selected by all the groups. The implementation of the livelihood empowerment category would require 32% of the total stickers (budget) and all groups selected these interventions without any trade off (Table 3).

Participants mentioned that the implementation of livelihood empowerment interventions would facilitate agricultural activities sand improve livelihood which will enhance incomes and access to diverse diets in their communities. Participants believed that the products from the farms would improve community nutritional intakes and that the income generated would enable community members to buy nutritious foods that they could not cultivate on their farms. Also, participants mentioned that livelihood skills training such as dress making would create additional jobs for community members and therefore help them to generate income.

> *"The reason why we say the farming is that, here we don't have any form of work doing aside the farming. The farming is our strength. Our children that are in school, it is the farming that helps us to take care of our children. When the child goes to school and returns home without eating it is not good. The child sits in the class and the stomach is aching. So, we know the farming is our strength" (Women's group, 40–50)*

> *"When we get water and farm inputs, it will enable us to improve our farming activities given that it is the thing(agriculture) that most of us do here to survive. When this is done the other progammes can now be considered" (Men's group, 40–50 years).*

> *"What I want to say is that, in terms of seedlings, if we could get seedlings, because now, some seedlings are there, we don't have them here. So you the health workers if you could help us to get those seedlings to grow" (Men's group, 24–34 years)*

*Micronutrient supplementation*. The nutrition-specific interventions that were selected by all 6 groups were the micronutrient supplementation interventions which included iron-folate supplementation and food fortification. The implementation of micronutrient supplementation interventions would require 45% of the total budget (Table 3).

The reason given by participants for the choice of these interventions reflected the poverty in the area which led to the community being poorly nourished. Participants explained that if community members were provided with fortified food and food supplements (iron folate supplementation), these would improve their well being and their overall nutritional status. Study participants further explained that the foods which currently make-up the majority of community members' diet do not contain the nutrients they require, explaining why their children and pregnant women are usually anaemic.

> *"In this community we are poor. Getting good food to eat is a problem. So when we are given food that is nutritious, it will help in our health and development. We don't have good foods in our houses and it is the reason why the health workers are saying that our children and pregnant women are malnourished (aneamic)" (Women group, 26–39 years)*

> *"Even though we get some food to each, though not enough, we know that if we get some vitamins or nutritious foods, particularly for pregnant women and children, it would help them a lot" (Men group, 40–50 years)*

*Male involvement*. The other prioritized nutrition-sensitive intervention was male involvement. This involved organizing men into groups and engaging them in nutrition education in order to improve nutrition intake in their households. Out of the 6 groups, only one group of women aged between 35 and 50 years (Group 1) did not prioritize male involvement as an

intervention that would improve the nutrition of the community (Table 3). The other 5 groups recognized the importance of male involvement and they explained that men are usually the household heads and the main decision makers in the household and they are respected as such. Therefore, engaging men through education about the importance of good nutrition and making them responsible would improve the dietary intake of the household.

"*Nowadays our men don't understand things. When you are pregnant, he doesn't pay attention to you. He is not bothered to find out as you are pregnant how you are doing. He doesn't have time for you. If he is present in the meetings and they talk and he listens, when you are pregnant and there is a problem, for instance the foods they say we should eat, he can say 'oh! these are the foods they said I should give to my wife. Let me search and get these foods for my wife to eat so that when she gives birth, the baby will be healthy.' That is why we chose that*" *(Women group, 40–50 years)*

*"If you look at intervention three, male involvement in nutrition education, it is important because it's not all of us (we the men) who will know what to do or what ingredients is needed to cook nutritious food for the family. If we can get people to sit with us continuously and educate us well on nutrition it will also help in our lives" (Men group, 35–50)*

*Less prioritized interventions.* The nutrition education programmes that were less prioritised were community nutrition education and youth nutrition education interventions. Participants understood that these programmes were likely to have no direct benefit unlike those interventions which provided food or increased income. Though participants did feel that these programmes were also important, because they had a limited number of stickers they had to choose and therefore prioritized other interventions which had more direct benefits than educational interventions.

*"Though the other interventions are equally important, our stickers are not much, and we need to choose the most important ones. The other interventions like educations can only go well for us if we are satisfied, if we get enough food and have enough money.*

*(Men group, 40–50)"*

Furthermore, participants reported a number of challenges that limited the use of nutrition education via SMS and radio. This included participants' inability to read text messages and lack of money to buy phones and radios on which to receive information. With all challenges considered, the lack of priority given to these interventions was not surprising.

*"Why we did not take the SMS and posters is because a lot of us have not been to school so if we receive text messages, and since we have not been to school, we will not be able to read the message, that is why we did not choose the SMS and posters (Women group, 26–39)"*

## Participants' impressions of the group deliberations

Participants were asked about how they had experienced the CHAT exercise, including how they had utilized the CHAT materials, how negotiations took place within the group and how they were able to work collectively as a group to prioritize interventions. All participants said that they had been excited to take part in the CHAT exercise. They felt working together in a group and the deliberations over prioritizing the interventions had gone smoothly and were educative, and that choosing priorities by consensus was cordial. The fact that despite

participants being drawn from different households across the community they were able to deliberate and to attain consensus was felt to be a great achievement that they had not initially believed was possible. It was clear from observation that participants debated over which interventions to choose before settling on a particular intervention. Though many participants had limited literacy, the explanations from the facilitators and the pictures used to illustrate the interventions enabled the CHAT processes, deliberations and the selection of interventions.

Participants perceived the CHAT exercise to be successful, the intervention options to be relevant and appropriate for the communities, and they were satisfied with the choices they made. They said that they would be very happy if the prioritized interventions were to be implemented in the community.

> *"Coming together to work is very interesting to us. All of us are from this community but we don't meet each other to exchange greetings, but as we have come together here, some people said things that are pleasant and interesting. If we can always come together this way and discuss issues together or work together it will help the community as a whole (Men group, 35–40 years)"*

> *"Working together in this group we did not have any disagreement among ourselves and I see it to be a good thing because if we had quarreled, we would not have been able to work together as a group"(Women group,40–50 years).*

> *"To me, all we did here went on well, whether educated or not, we were able to understand the process, and whatever we have discussed here about nutrition also went well. If we can get what we chose as our most pressing needs implemented in the community, we will be very happy and grateful" (Women group, 26–39 years)*

## Discussion

This study sought to use CHAT to identify community members' priorities for nutrition interventions in rural communities in northern Ghana and the reasons for those priorities.

The nutrition-sensitive intervention that was prioritized over all others was the livelihood intervention. This was not unexpected given the poverty in the area and the main occupation in the community which is subsistence agriculture. In general, the communities are repeatedly confronted with erratic and inadequate rainfall which results in poor harvests leading to poverty and food insecurity. Findings from conversations with people in the same area confirm that poverty, lack of irrigated agricultural land and poor harvests are perceived to be the main barriers to optimal nutrition [15]. The seasonal effect of food insecurity has been shown to lead to higher risk of severe acute malnutrition among children aged less than three years old [19–21]. A previous study conducted in the study area revealed that being born in the 'hungry' season is associated with a higher risk of severe acute malnutrition [21]. Hence the priority given by the community to improving the food supply to pregnant and lactating mothers through livelihood empowerment interventions and sustainable agriculture specifically to reduce the burden of severe acute malnutrition.

Community members in this region are mainly subsistence farmers. They are not able to engage in farming activities during the dry season and the produce and income usually obtained during the rainy seasons are often insufficient. This is one reason for the poverty in the region. The Upper East region, where this study was carried out, is the second poorest area in Ghana with poverty rate of 44.4% [22]. It is therefore understandable that participants prioritized interventions that have the potential to improve agricultural and other livelihood

activities in the area. To improve nutritional intakes in such communities, policy makers should provide a regular supply of water for all year-round agricultural activities, improve livelihood through income generating activities and skills training in dress making, soap making etc. The current government (New Patriotic Party) has identified this problem and as part of its election campaign, promised that in each village in northern Ghana, a dam will be constructed to boost agricultural activities [23].

The other prioritized nutrition-specific interventions were iron-folate supplementation and fortified foods for the reason that poverty is endemic in the study area and obtaining adequate nutritional foods for good health is a serious challenge. Commonly, in Ghana, deficiencies in micronutrients are related to inadequate intake of iron, iodine, folic acid and other vitamins [24, 25]. Therefore, iron and folic acid supplementation and food fortification would reduce the anaemia burden among vulnerable groups [25]. The community members were well aware of this and, as a consequence, prioritized these interventions. A previous study showed that long-term routine use of micronutrient powder containing prophylactic iron reduced anaemia, iron deficiency and iron deficiency anaemia among pre-school children living in rural Ghana [26]. A recent Lancet review confirmed that micronutrient supplementation was an effective intervention to improve maternal and child nutrition [27].

Male involvement was another priority nutrition-sensitive educational intervention for all the study groups. In a typical patriarchal society like that which exists in the study area, male involvement is key to promoting nutritional health in the community. When men are involved in any activity in the household, it makes them responsible and the outcomes improve [28]. As the study participants observe, when men are involved they will appreciate the importance of good nutrition and are more likely to support the acquisition of nutritional foods for their households, particularly for pregnant women and children [29]. The association of male involvement with improved nutritional status of the children has been shown in other studies [25, 28, 29]. In a study conducted in Ghana on male involvement in maternal healthcare, it was concluded that sustained male involvement, particularly of husbands, is required at the household and community levels for positive maternal outcomes [30]. This therefore suggests that male involvement in nutrition should be given serious consideration by those involved in efforts to improve nutrition and to reduce malnutrition.

The study showed that the participants were able to work together to select interventions that they considered a priority in improving the nutrition of mothers and children in the community, and were able to give appropriate reasons for their choices. Participants were able to make selections considering the cost of implementation of interventions and to make trade-offs. These findings are consistent with previous CHAT studies that reported smooth deliberations in the selection of health insurance benefit packages [3, 4, 7]. Despite the low literacy level in the community, participants were able to use the CHAT tools and supporting materials in order to prioritize nutrition interventions. The study revealed that participants were happy working together in the deliberations to prioritize interventions that would benefit their community members.

The study supports the importance of community engagement in priority decisions to improve services in the communities. Policies are usually developed by those working at some distance from the communities in which they are intended to be implemented, a situation which can generate implementation challenges. A systematic review was conducted to evaluate the effectiveness of public health interventions that engage the community on a range of health outcomes across diverse health issues and they concluded that there is solid evidence that community engagement interventions have a positive impact on a range of health outcomes across various conditions [31]. A typical example of the difference that community involvement can make in health care is the Ghana CHPS programme. Community involvement in planning,

implementing, and evaluating CHPS activities has been reported to have improved health and programme outcomes [32, 33].

## Strengths and limitations of the study

One important strength of this study is that the process of identifying the prevailing nutrition interventions was robust. It included the triangulation of evidence from lay and published literature, interaction with different stakeholders and the involvement of the community.

The second strength is the use of lay community members to pilot CHAT and prioritise nutrition interventions in the face of diverse intervention options with limited resources. This gave the opportunity for community members to appreciate the processes involved in prioritising and implementing health interventions. The implementation process also gave insights to the community members of public deliberation exercise and overall the insights on setting priorities in intervention. It might also lead to a sense of appreciation among community members and improved adherence to existing interventions haven appreciated the associated implementation cost. The successful adoption, modification and pilot implementation of CHAT for nutrition interventions and in a rural setting is a major strength to our study. It implies that, the methodology used can be scaled up to engage more groups and replicated in other settings in Ghana. The tool perhaps may be modified and used to prioritise other health related outcomes beyond nutrition. This premier study has generated reliable results and may serve as a benchmark for future CHAT related studies in Ghana and elsewhere in sub-Saharan Africa.

Despite these strengths, we would like to acknowledge a few limitations to our study. First, the lengthy time commitment from participants to undertake the CHAT exercises. This was a considerable inconvenience to some participants since it took an average of 3 hours. Given that the participants could not read the instructions, a lot of time was spent by the facilitators in explaining the exercise to participants in the local language. Nevertheless, participants appeared to find the CHAT sessions interactive and intriguing as facilitators were able to maintain the focus of the participants throughout the exercise.

CHAT instructions were translated from English into the local language. It is possible that the intended meaning of some of the instructions may have been lost during the translation process. However, the facilitators were experienced research officers who are natives to the area, thus minimizing the potential for mistranslations making this unlikely to affect the study findings.

Another limitation of the study is the small number of groups (6 groups) used for the pilot implementation of the CHAT exercise which we believe may not be a representative of the studied population. This may limit the generalizability of our findings to the study population.

## Conclusions

The study showed that it is feasible to use CHAT in the prioritization of nutrition intervention programmes in rural settings in communities with low levels of literacy. The CHAT exercise provided insight into communities' nutrition priorities and identified livelihood empowerment, micronutrient supplementation and male involvement in household nutrition as key choices in the study community. The communities gave clear justifications for their choices based on their community needs and shared experiences. There is independent evidence that these prioritized intervention programmes when implemented will improve nutritional intake and help reduce micro- and macro-nutrient deficiencies. We strongly recommend CHAT as a participatory approach to priority setting for nutrition interventions in Ghana and in other low- and middle-income countries. We further propose that future research should focus on the use of CHAT for other types of health interventions and in different populations. We also

recommend a comparative study to examine any differences or similarities in priorities identified by community members and stakeholders and the reasons supporting the choices that would be made.

## Supporting information

**S1 Table. Policy documents reviewed.**
(DOCX)

**S2 Table. CHAT manual.**
(DOCX)

**S1 Text. Facilitator script.**
(DOCX)

**S2 Text. Nutrition scenario cards.**
(DOCX)

## Acknowledgments

The authors acknowledge with gratitude all the support and contributions from the various institutions and individuals. Specifically, we would like to acknowledge the Navrongo Health Research Centre for the institutional support to undertake the study.

We acknowledge the data collectors for their contribution to data collection in the field. We are also thankful to the study participants for their time and contributions to the study.

We also salute the intellectual contributions of the entire INPreP study group particularly work package 4 members for their support in the study.

## Author Contributions

**Conceptualization:** Maxwell Ayindenaba Dalaba, Engelbert A. Nonterah, Samuel T. Chatio, James K. Adoctor, Edith Dambayi, Stephen Azalia, Doreen Ayi-Bisah, Agnes Erzse, Daniella Watson, Polly Hardy-Johnson, Sarah H. Kehoe, Aviva Tugendhaft, Kate Ward, Cornelius Debpuur, Abraham Oduro, Winfred Ofosu, Marion Danis, Mary Barker.

**Data curation:** Maxwell Ayindenaba Dalaba, Engelbert A. Nonterah, James K. Adoctor, Edith Dambayi, Esmond W. Nonterah, Stephen Azalia, Doreen Ayi-Bisah, Agnes Erzse, Daniella Watson, Polly Hardy-Johnson, Sarah H. Kehoe, Cornelius Debpuur.

**Formal analysis:** Maxwell Ayindenaba Dalaba, Engelbert A. Nonterah, James K. Adoctor, Edith Dambayi, Doreen Ayi-Bisah, Agnes Erzse, Daniella Watson, Polly Hardy-Johnson, Sarah H. Kehoe, Aviva Tugendhaft, Kate Ward, Cornelius Debpuur, Winfred Ofosu, Marion Danis, Mary Barker.

**Investigation:** Engelbert A. Nonterah.

**Methodology:** Maxwell Ayindenaba Dalaba, Engelbert A. Nonterah, Samuel T. Chatio, James K. Adoctor, Edith Dambayi, Esmond W. Nonterah, Doreen Ayi-Bisah, Agnes Erzse, Daniella Watson, Polly Hardy-Johnson, Sarah H. Kehoe, Aviva Tugendhaft, Cornelius Debpuur, Abraham Oduro, Winfred Ofosu, Marion Danis, Mary Barker.

**Project administration:** Engelbert A. Nonterah.

**Resources:** Abraham Oduro, Mary Barker.

**Supervision:** Maxwell Ayindenaba Dalaba, Engelbert A. Nonterah, Samuel T. Chatio, James K. Adoctor, Edith Dambayi, Esmond W. Nonterah, Stephen Azalia, Doreen Ayi-Bisah, Sarah H. Kehoe.

**Validation:** Maxwell Ayindenaba Dalaba, Engelbert A. Nonterah, Stephen Azalia, Doreen Ayi-Bisah, Daniella Watson, Polly Hardy-Johnson, Sarah H. Kehoe, Aviva Tugendhaft, Kate Ward, Cornelius Debpuur, Abraham Oduro, Winfred Ofosu, Marion Danis, Mary Barker.

**Visualization:** Maxwell Ayindenaba Dalaba, Engelbert A. Nonterah, Samuel T. Chatio, James K. Adoctor, Edith Dambayi, Esmond W. Nonterah, Stephen Azalia, Doreen Ayi-Bisah, Agnes Erzse, Daniella Watson, Polly Hardy-Johnson, Sarah H. Kehoe, Aviva Tugendhaft, Kate Ward, Cornelius Debpuur, Abraham Oduro, Winfred Ofosu, Marion Danis, Mary Barker.

**Writing – original draft:** Maxwell Ayindenaba Dalaba, Engelbert A. Nonterah, James K. Adoctor, Edith Dambayi, Esmond W. Nonterah, Stephen Azalia, Doreen Ayi-Bisah, Agnes Erzse, Daniella Watson, Polly Hardy-Johnson, Sarah H. Kehoe, Aviva Tugendhaft, Kate Ward, Cornelius Debpuur, Abraham Oduro, Winfred Ofosu, Marion Danis, Mary Barker.

**Writing – review & editing:** Maxwell Ayindenaba Dalaba, Engelbert A. Nonterah, Samuel T. Chatio, James K. Adoctor, Edith Dambayi, Esmond W. Nonterah, Stephen Azalia, Doreen Ayi-Bisah, Agnes Erzse, Daniella Watson, Polly Hardy-Johnson, Sarah H. Kehoe, Aviva Tugendhaft, Kate Ward, Cornelius Debpuur, Abraham Oduro, Winfred Ofosu, Marion Danis, Mary Barker.

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
