## [Decision Letter · Decision Letter 0]

11 Jan 2022

PGPH-D-21-00821

Engaging community members in priority setting for nutrition interventions in rural northern Ghana

Dear Dr. Dakaba,

Thank you for submitting your manuscript to PLOS Global Public Health. After careful consideration, we feel that it has merit but does not fully meet PLOS Global Public Health’s publication criteria as it currently stands. Therefore, we invite you to submit a revised version of the manuscript that addresses the points raised during the review process.

We look forward to receiving your revised manuscript.

Kind regards,

Jitendra Kumar Singh, PhD

Academic Editor

Journal Requirements:

1. Please provide separate figure files in .tif or .eps format only, and remove any figures embedded in your manuscript file.  If you are using LaTeX, you do not need to remove embedded figures.

For more information about figure files please see our guidelines: https://journals.plos.org/globalpublichealth/s/figures

2. Please amend your detailed Financial Disclosure statement. This is published with the article, therefore should be completed in full sentences and contain the exact wording you wish to be published.

i) Please include all sources of funding (financial or material support) for your study. List the grants (with grant number) or organizations (with url) that supported your study, including funding received from your institution. 

ii). State the initials, alongside each funding source, of each author to receive each grant.

iii). State what role the funders took in the study. If the funders had no role in your study, please state: “The funders had no role in study design, data collection and analysis, decision to publish, or preparation of the manuscript.”

Additional Editor Comments (if provided):

Reviewers' comments:

Reviewer's Responses to Questions

**Comments to the Author**

1. Does this manuscript meet PLOS Global Public Health’s publication criteria? Is the manuscript technically sound, and do the data support the conclusions? The manuscript must describe methodologically and ethically rigorous research with conclusions that are appropriately drawn based on the data presented.

Reviewer #1: Yes

Reviewer #2: Yes

2. Has the statistical analysis been performed appropriately and rigorously?

Reviewer #1: Yes

Reviewer #2: Yes

3. Have the authors made all data underlying the findings in their manuscript fully available (please refer to the Data Availability Statement at the start of the manuscript PDF file)?

Reviewer #1: Yes

Reviewer #2: Yes

4. Is the manuscript presented in an intelligible fashion and written in standard English?

Reviewer #1: Yes

Reviewer #2: Yes

5. Review Comments to the Author

Reviewer #1: Reviewer’s comments

Manuscript ID PGPH-D-21-00821: Engaging community members in priority setting for nutrition interventions in rural northern Ghana

General comments

Above mentioned descriptive cross-sectional study perfectly describes that priority setting for ‘nutrition interventions’ is possible and feasible by engaging community members with the aid of a simulation decision-making tool namely ‘Choosing All Together’ (CHAT) in a rural setting of low literacy and resources. The manuscript is well organized, so is the methodology which can be adopted in resource-poor setting with higher malnutrition prevalence, where prioritizing is important for funding in the area of nutrition health policy strategy and implementation. However, some minor unintended errors (especially numbering references in the text) are observed in some places of the manuscript. I strongly recommend to accept this paper with some minor corrections.

Specific comments

Introduction

Between line 100 and 101: Instead of giving reference number 3 (Tugendhaft et al. 2021) authors (with colleagues) names are written; however, together with Tugendhaft et al. 2021, name of Hurst et al. 2018 (wrongly?) is also mentioned which has reference number 7 in reference list, but serially next references are 4, 5 (in the line 104), and then references 6, 7 are in the line 111 (along with reference 3).

Authors carefully should resolve above mentioned problems along with reference list.

Material and Methods

Study area

Between line 130 and 131: Instead of giving reference number 10 (Dalaba et al. 2016) and 11 (Oduro et al. 2012), authors (with colleagues) names are written without closing round bracket like ---Dalaba etal. 2016; Oduro et al. 2012).

Study population and sampling

Line 153: Should elaborate first e.g. In depth interviews (IDIs), though it is elaborated and mentioned later in the line 195.

Results

Socio-demographic characteristics of participants

Line 305: table 1 should also be mentioned in the parenthesis as table 2 has no age related data

Discussion

Line 497: after the line ‘constructed to boost agricultural activities when voted into power’ insertion of number (30) (reference?) does not make any sense as reference 30 does not exist in reference list.

Reviewer #2: In this study, which aims to identify priorities in nutrition interventions and the reasons for choices in communities living in Northern Ghana, what makes the study important is that it supports the importance of community participation in priority decisions to improve services in communities.

Below you can find some arrangements for organizing the writing of the work specifically to improve the materials and methods, results and discussion.

INTRODUCTION

The citations on lines 100-101 should be corrected and reviewed in the references accordingly.

The introduction should be expanded by briefly mentioning similar practices in other countries.

MATERIAL AND METHOD

The citations on lines 130-131 should be corrected and should be reviewed in the same way.

It should be defined where the abbreviations are first mentioned and should be used optionally as abbreviations in the following sections. (such as FGD and IDI abbreviations)

Other abbreviations should be checked.

DISCUSSION

484-485 lines. This section should be expanded by scanning the literature for different age groups in the lines.

488-490 lines should be referenced.

500-506 lines should be expanded with studies on this subject, especially based on the current literature.

507-517 lines studies on male participation should be added and expanded.

Limitations of the study should be added.

It is recommended to mention the strengths of the study and suggestions for future research in the conclusion.

REFERENCES

It should be reviewed by paying attention to the abbreviations in the journal names (references such as 13, 14, 15)

There appear to be two 29th sources, should be checked.

6. PLOS authors have the option to publish the peer review history of their article (what does this mean?). If published, this will include your full peer review and any attached files.

**Do you want your identity to be public for this peer review?** For information about this choice, including consent withdrawal, please see our Privacy Policy.

Reviewer #1: **Yes: **Mahbuba Kawser

Reviewer #2: No

---

## [Editor Report · Decision Letter 1]

16 Feb 2022

PGPH-D-21-00821

Engaging community members in priority setting for nutrition interventions in rural northern Ghana

Dear Dr. Dakaba,

Thank you for submitting your manuscript to PLOS Global Public Health. After careful consideration, we feel that it has merit but does not fully meet PLOS Global Public Health’s publication criteria as it currently stands. Therefore, we invite you to submit a revised version of the manuscript that addresses the points raised during the review process.

A rebuttal letter that responds to each point raised by the editor and reviewer(s). You should upload this letter as a separate file labeled 'Response to Reviewers'.

We look forward to receiving your revised manuscript.

Kind regards,

Jitendra Kumar Singh, PhD

Academic Editor

Journal Requirements:

1. Please amend your detailed Financial Disclosure statement. This is published with the article, therefore should be completed in full sentences and contain the exact wording you wish to be published.

i). Please include all sources of funding (financial or material support) for your study. List the grants (with grant number) or organizations (with url) that supported your study, including funding received from your institution. 

ii). State the initials, alongside each funding source, of each author to receive each grant.

iii). State what role the funders took in the study. If the funders had no role in your study, please state: “The funders had no role in study design, data collection and analysis, decision to publish, or preparation of the manuscript.”

2. Please ensure that the funders and grant numbers match between the Financial Disclosure field and the Funding Information tab in your submission form. Note that the funders must be provided in the same order in both places as well.

3. The resolution of Figure 1 is very low and somewhat difficult to read. It is important that our Editors and Peer Reviewers are able to read all parts of a submission. Please replace this figure with higher resolution copies.

Please provide separate figure files in .tif or .eps format only and ensure that all files are under our size limit of 20MB.

Reviewers' comments:

Reviewer's Responses to Questions

Comments to the Author

1. Does this manuscript meet PLOS Global Public Health’s publication criteria? Is the manuscript technically sound, and do the data support the conclusions? The manuscript must describe methodologically and ethically rigorous research with conclusions that are appropriately drawn based on the data presented.

Reviewer #1: Yes

Reviewer #2: Yes

2. Has the statistical analysis been performed appropriately and rigorously?

Reviewer #1: Yes

Reviewer #2: Yes

3. Have the authors made all data underlying the findings in their manuscript fully available (please refer to the Data Availability Statement at the start of the manuscript PDF file)?

Reviewer #1: Yes

Reviewer #2: Yes

4. Is the manuscript presented in an intelligible fashion and written in standard English?

Reviewer #1: Yes

Reviewer #2: Yes

5. Review Comments to the Author

Reviewer #1: Reviewer’s comments

Manuscript ID PGPH-D-21-00821: Engaging community members in priority setting for nutrition interventions in rural northern Ghana

General comments

Above mentioned descriptive cross-sectional study perfectly describes that priority setting for ‘nutrition interventions’ is possible and feasible by engaging community members with the aid of a simulation decision-making tool namely ‘Choosing All Together’ (CHAT) in a rural setting of low literacy and resources. The manuscript is well organized, so is the methodology which can be adopted in resource-poor setting with higher malnutrition prevalence, where prioritizing is important for funding in the area of nutrition health policy strategy and implementation. However, some minor unintended errors (especially numbering references in the text) are observed in some places of the manuscript. I strongly recommend to accept this paper with some minor corrections.

Specific comments

Introduction

Between line 100 and 101: Instead of giving reference number 3 (Tugendhaft et al. 2021) authors (with colleagues) names are written; however, together with Tugendhaft et al. 2021, name of Hurst et al. 2018 (wrongly?) is also mentioned which has reference number 7 in reference list, but serially next references are 4, 5 (in the line 104), and then references 6, 7 are in the line 111 (along with reference 3).

Authors carefully should resolve above mentioned problems along with reference list.

Material and Methods

Study area

Between line 130 and 131: Instead of giving reference number 10 (Dalaba et al. 2016) and 11 (Oduro et al. 2012), authors (with colleagues) names are written without closing round bracket like ---Dalaba etal. 2016; Oduro et al. 2012).

Study population and sampling

Line 153: Should elaborate first e.g. In depth interviews (IDIs), though it is elaborated and mentioned later in the line 195.

Results

Socio-demographic characteristics of participants

Line 305: table 1 should also be mentioned in the parenthesis as table 2 has no age related data

Discussion

Line 497: after the line ‘constructed to boost agricultural activities when voted into power’ insertion of number (30) (reference?) does not make any sense as reference 30 does not exist in reference list.

Reviewer #2: In this study, which aims to identify priorities in nutrition interventions and the reasons for choices in communities living in Northern Ghana, what makes the study important is that it supports the importance of community participation in priority decisions to improve services in communities.

Below you can find some arrangements for organizing the writing of the work specifically to improve the materials and methods, results and discussion.

INTRODUCTION

The citations on lines 100-101 should be corrected and reviewed in the references accordingly.

The introduction should be expanded by briefly mentioning similar practices in other countries.

MATERIAL AND METHOD

The citations on lines 130-131 should be corrected and should be reviewed in the same way.

It should be defined where the abbreviations are first mentioned and should be used optionally as abbreviations in the following sections. (such as FGD and IDI abbreviations)

Other abbreviations should be checked.

DISCUSSION

484-485 lines. This section should be expanded by scanning the literature for different age groups in the lines.

488-490 lines should be referenced.

500-506 lines should be expanded with studies on this subject, especially based on the current literature.

507-517 lines studies on male participation should be added and expanded.

Limitations of the study should be added.

It is recommended to mention the strengths of the study and suggestions for future research in the conclusion.

REFERENCES

It should be reviewed by paying attention to the abbreviations in the journal names (references such as 13, 14, 15)

There appear to be two 29th sources, should be checked.
---

## [Editor Report · Decision Letter 2]

13 Apr 2022

Engaging community members in setting priorities for nutrition interventions in rural northern Ghana

PGPH-D-21-00821R2

Dear Dr. Dalaba,

We are pleased to inform you that your manuscript 'Engaging community members in setting priorities for nutrition interventions in rural northern Ghana' has been provisionally accepted for publication in PLOS Global Public Health.

Best regards,

Jitendra Kumar Singh, PhD

Academic Editor